The larva and female of Perigomphus basicornis Amaya-Vallejo, Novelo-Gutiérrez & Realpe, 2017, and the first record of Perigomphus pallidistylus (Belle, 1972) for Colombia (Insecta: Odonata: Gomphidae)

Amaya-Vallejo Vanessa v.amaya10@uniandes.edu.co 1
Novelo-Gutiérrez Rodolfo 2
Realpe Emilio 1
1 Laboratorio de Zoología y Ecología Acuática LAZOEA, Universidad de Los Andes , Bogotá , Cundinamarca , Colombia
2 Instituto de Ecología A. C., Red de Biodiversidad y Sistemática , Xalapa , Veracruz , México
Esteban María Ángeles
Electronic publication date: 2018 Aug 8
Publication date: 2018
Volume: 6
Electronic Location ID: e5279
Received 2018 Mar 9; Accepted 2018 Jul 2
Copyright: ©2018 Amaya-Vallejo et al.
Copyright year: 2018
Copyright holder: Amaya-Vallejo et al.
License: This is an open access article distributed under the terms of the Creative Commons Attribution License, which permits unrestricted use, distribution, reproduction and adaptation in any medium and for any purpose provided that it is properly attributed. For attribution, the original author(s), title, publication source (PeerJ) and either DOI or URL of the article must be cited.
License URL: https://creativecommons.org/licenses/by/4.0/

Keywords: Larva, Taxonomy, Anisoptera, Colombia, Description, Female, Valle del Cauca, Caldas, Rearing, Habitat

Funding: UNESCO/KEIZO OBUCHI RESEARCH FELLOWSHIPS PROGRAMME (cycle 2010) Universidad de Los Andes Proyecto Semilla Grant 2016-II This work was supported by the UNESCO/KEIZO OBUCHI RESEARCH FELLOWSHIPS PROGRAMME (cycle 2010) and the Universidad de Los Andes Proyecto Semilla Grant (2016-II). The funders had no role in study design, data collection and analysis, decision to publish, or preparation of the manuscript.

==============================
The larva and female of Perigomphus basicornis are described and illustrated, and compared with the larva and female of P. pallidistylus. The larva of P. basicornis differs from that of P. pallidistylus in having sternum 8 divided in five sclerites, abdominal segments 8 and 9 with small, low protuberances on the tergites and male’s epiproct as long as its basal width, mainly. The female of P. basicornis differs from that of P. pallidistylus in having the apical lobes of vulvar lamina wider, with divergent tips. P. pallidistylus is recorded for Colombia for the first time.

Introduction

The Gomphidae family is a well-studied group comprising the largest number of genera and species next to the Libellulidae; it contains about 941 spp. in 92 genera (Garrison, Ellenrieder & Louton, 2006) which can be found across North and South America, Europe, Asia, and Australia. In the New World, 355spp. and 34 genera have been described, being 31 of them endemic (Garrison, Ellenrieder & Louton, 2006). Also, most of these Gomphids are associated with pristine environments due to their biotope requirements (Corbet, 1999), showing the high potential of the Neotropical Forests as biodiversity reservoirs. These species are not resilient to changes in habitat structure, so deforestation and degradation of water bodies notoriously affects the species composition inside a determined zone. For this reason, description of new species and their classification inside IUCN concern categories are of prior importance inside Odonatology.

The Neotropical dragonfly genus Perigomphus remained as monotypic for almost 40 years since Belle (1972) made the original description of P. pallidistylus, native from Costa Rica and Panama (Garrison, Ellenrieder & Louton, 2006). Actually, there are two more described species based only on male specimens: P. angularis (Tennessen, 2011), known from the Amazon headwaters of central Ecuador, and P. basicornis (Amaya-Vallejo, Novelo-Gutiérrez & Realpe, 2017), endemic to the tropical rainforests of Anchicayá, Valle del Cauca department, Colombia. No larva or females had been described, only for P. pallidistylus (Westfall, 1989). Here we describe the male and female larva of P. basicornis from exuviae and mature larvae, as well as the female imago from specimens reared under laboratory conditions. Larvae of P. pallidistylus were collected in the Department of Caldas, becoming the first register for the species in Colombia.

Figure 1 Map of the sampling zone.

(A) Location of the Anchicayá River Basin inside Valle del Cauca Departament, Colombia (in green). (B) Location of ALBAN inside Anchicayá River Basin (red polygon). Map from CVC-UV (2007).

Perigomphus appears to be an elusive genus, with adult individuals inhabiting the high canopy and just descending for mating. The original description of the genus (Belle, 1972) was made based upon only two male specimens from Costa Rica, one of them a young adult without penis. Westfall (1989) described the larva and the female of P. pallidistylus from an emerging specimen collected in field along with its exuvia, but all the paratypes (two males and four females) were reared under lab conditions. Larvae seem to be easier to find and collect, as the description material includes 43 specimens from Costa Rica, but no adults. No remarks about the habitat preference of the species were made on these articles. The description made of the second species is based on a solitary male specimen, found in a small, slow-flowing, shallow stream at 820 masl, with a bottom composed of silt, sand, and rock. It was perched in a sunny spot amid small trees, on the tip of a twig about 1 m above ground, in the early afternoon (Tennessen, 2011), so mating activity can be inferred from this. But no other material is available.

For P. basicornis, the situation is similar. Along 5 years of sampling, not a single adult was collected, although a considerable amount of larvae resembling the one of P. pallidistylus were found in pristine forest streams. Therefore, it was hypothesized that this could be a new species according to some spotted morphological differences, but the presence of an adult and the life stages correspondence were necessary to have a completely reliable description. Knowing the biotic and abiotic variables (Amaya-Vallejo & Realpe, 2007) preferred for the larvae of this possible new species, an artificial habitat was built under lab conditions and after two years, enough material was obtained for an accurate larva-adult association and new species description. Original description was made only for males, as finding, rearing and obtaining enough material for females took almost an additional year.

Materials & Methods

The sampling zone is located in the Anchicayá River Basin, Valle del Cauca Departament, Colombia (Fig. 1A), inside the perimeter of the High and Low Anchicayá Hydroelectric Central Complex (ALBAN, Spanish abbv.) (Fig. 1B). ALBAN is covered with Tropical Rainforests full of streams, creeks and rivers hosting a rich and diverse assemblage of Odonata (Amaya-Vallejo & Realpe, 2007). Previous work (Amaya-Vallejo & Novelo-Gutiérrez, 2011; Amaya-Vallejo & Novelo-Gutiérrez, 2012; Amaya-Vallejo, Novelo-Gutiérrez & Realpe, 2017; Bota-Sierra, Novelo-Gutiérrez & Amaya-Vallejo, 2017) had shown that sampling inside ALBAN increases the chances of finding new species, as it is a very well conserved zone. The temperatures range between 30°–21 °C with an average of 28 °C and 86% humidity all year long. Larvae were collected in four different field trips between 2016 and 2017 from pristine forest creeks and streams showing the required conditions for the species to inhabit them (Amaya-Vallejo, Novelo-Gutiérrez & Realpe, 2017). They were retrieved with D-nets and the mature ones were kept alive to be reared under laboratory conditions, in an artificial habitat built following DuBois & Tennessen (2016) guidelines but with modifications according to larvae environmental requirements (Amaya-Vallejo & Realpe, 2007) (Table 1) and behavior observed in situ. The habitat was kept inside a warm room emulating the original environmental temperature, which maintained the water temperature at the ideal levels. The high levels of water oxygen were preserved with three generic aquarium pumps, and the components of the bottom were taken directly from the natural habitats. Water level was replenished every two days with clean tap water. Larvae were fed three times per week with Grindall worm, Enchytraeus buchholzi.

Table 1 Standardized variables for Perigomphus basicornis rearing under laboratory conditions.

Variable	Ideal values	
Substrate of bottom	Coarse sand, pebbles, rocks, leaflitter.	
Surrounding temperature	24 °C–25 °C.	
Water temperature	20 °C–24 °C	
Dissolved oxygen	10–12 mg/L	

Five were successfully reared until emergence; they took 2–8 months to emerge. Tenerals were put inside a box (DuBois & Tennessen, 2016) to let them harden. Then they were killed with cold and stored in envelopes. Eleven larvae died without rearing and were preserved in 96% ethanol.

An emerging specimen was collected in field, along with its exuvia.

Field Research was approved by Parques Naturales Nacionales de Colombia (PNN, Spanish abbreviation) office, Permit Number 005 of 2016.

Photographs of morphology were taken with a Nikon DS-U3 camera mounted on a stereomicroscope Nikon SMZ25, and processed with the program NIS elements AR version 4.5, images focus stacked. Descriptions were made under a stereomicroscope Zeiss Stemi SV6, and measurements (in mm) taken with an ocular micrometer and a ruler. Wing nomenclature follows Riek & Kukalová-Peck (1984). Mandible nomenclature follows Watson (1956); labium nomenclature follows Corbet (1953). Abbreviations are as follows: AL, abdomen length; FwL, Forewing length; HwL, hindwing length; HfL, hind femur length; MWh, maximum width of head; Pt, pterostigma, CL, cerci length; S1–10, abdominal segments; TL, total length (including caudal appendages); IEXA, Colección Entomológica “Miguel Angel Morón Ríos” from Instituto de Ecología, A.C., Xalapa, and ANDES Entomology Museum, Universidad de Los Andes, Bogotá. Specimens are deposited at ANDES (holotype) and IEXA (allotypes). Larval development stages nomenclature (F) as defined by Corbet (1999).

Results

Perigomphus basicornis Amaya-Vallejo, Novelo-Gutiérrez & Realpe	
(Figs. 2–7)	

Figure 2 Habitus of Perigomphus basicornis, F-0 larva: (A) dorsal; (B) ventral.

Photo by Rodolfo Novelo-Gutiérrez.

Material. Six exuviae (Three male, three females, reared), 11 F-0 larvae (three males, four females). COLOMBIA: Alto Anchicayá, Anchicayá River Hydroelectric Central (CHIDRAL, Spanish abb.), elevation 630 m; LB1creek (3.5771667N, −76.880666W), 5 July 2016 (three females, two males [ANDES:VAV2016:LB11]); La Esperanza Creek (36126667N, 76.88836111111112W), 6 September 2016 (one female reared under lab conditions: emerged 12 May 2017 [ID: ANDES:VAV2016:EZPA1], exuvia kept, imago used for paratype description); La Loquita creek (3.5903056N, −76.88869444W), 9 September 2016 (three males reared under lab conditions: emerged 22 November 2016 [IEXA], 24 March 2017 and 18 April 2017 [ID: ANDES.VAV2017:LQTA4]; 15 December 2017 (1 female reared under lab conditions: emerged 3 April 2018 [ID: ANDES.VAV2017:LQTA8]); La Riqueza River (3.6094167N, −76.8845W), 10 September 2016 (1 female emerging in situ, exuvia kept, imago used for holotype description [ID: ANDES:VAV2016:RIQ13]); all V. Amaya leg. Specimens deposited at ANDES and IEXA.

Figure 3 Details of the morphology of Perigomphus basicornis.

(A) Ventral view of head showing mouthparts (labium removed); (B) head, dorsal view; (C) right mandible; (D) left mandible, both in ventrointernal view. Photo by Rodolfo Novelo-Gutiérrez.

Figure 4 Prementum of Perigomphus basicornis:. (A) ventral; (B) dorsal.

Photo by Rodolfo Novelo-Gutiérrez.

Description of the F-0 larva. Small-size larva, body sturdy and entirely covered by minute scale-like setae giving it a granular aspect, with large, irregular, bare areas on occiput and pronotum; abdomen enlarged, parallel-sided, gently tapering caudad; body light brown to brown, lacking any particular color pattern (Fig. 2).

Head: Small, wider than long, narrower than thorax and abdomen (Fig. 2A). Labrum granulose, anterior border widely convex with an external row of long, white and grayish-white, stiff setae (Fig. 3A), flattened ventrally, ventrointernal margin concave, with a dense brush of long setae; anteclypeus bare; postclypeus, frons, vertex and occiput granulose; frons (Fig. 3B) slightly produced as a short shelf, with anterior margin slightly convex and anterolateral corners widely rounded, fronto-lateral margins of frons with a tuft of long, upturned golden setae; occiput granulose with large, bare, irregular areas, cephalic lobes bulging, occipital margin waiving. Antennae (Fig. 2A, Fig. 3B) 4-segmented, with abundant, minute, scale-like setae, scape and pedicel ring-like, 3rd segment the largest, plate-like, flattened dorso-ventrally, 0.10 longer than its widest part, in ventral view apical margin thick (Fig. 3A), with a series of long, stiff, upturned setae, in dorsal view (Fig. 3B) internal margin greatly expanded medially, external margin widely convex; 4th segment strongly reduced to a minute sphere; relative length of antenomeres: 0.2, 0.2, 1.0, 0.04. Compound eyes relatively small, ocelli white (Fig. 2A). Mandibles (Fig. 2C–Fig. 2D) with molar crest, mandibular formula: L 1234 0 a (m1,2,3,4or5,6)b /R 1234 y a(m1or2)b, in both mandibles tooth a >b. Maxillae: Galeolacinia (Fig. 3A) with seven moderately incurved, acute teeth; three dorsal teeth more or less of same length and robustness and four ventral teeth of different size, apical one the largest; maxillary palp thick and robust, setose. Ventral pad of hypopharynx (Fig. 3A) whitish, soft, anterior half covered with long, stiff setae, a pentagonal sclerite on basal half. Labium: prementum-postmentum articulation slightly surpassing the level of procoxae. Prementum (Fig. 4) reddish-brown to brown, subquadrate, maximum width-length ratio 0.96:1, lateral margins slightly serrulated, subparallel on apical half, moderately converging on basal half, basal margin straight, without a longitudinal, central sulcus on ventral surface (Fig. 4A); dorsal surface (Fig. 4B) with a lateral, sub-basal group of small tubercles beset with small spiniform setae. Ligula (Fig. 4) convex, moderately prominent, one third the length of its base, distal margin very slightly serrate with a dorsal row of long pilliform setae (Fig. 4B) and two short, stout, blunt teeth on the middle (Fig. 4A); dorsal surface of ligula abundantly covered with long, stiff setae, some of them as long as pilliform setae. Labial palp (Fig. 4) the same color than prementum on basal half, distal half darker, dorsal surface bare, ventral surface with some long, delicate, hair-like setae and very minute spiniform setae, ending in a stout, incurved, sharp tooth, internal margin with 8–9 incurved, sharply pointed teeth decreasing in size from the tip to the base, last almost vestigial; movable hook reddish-brown, shorter than palp, sharp and moderately incurved.

Figure 5 Details of the morphology of Perigomphus basicornis.

(A) right foreleg; (B) tergites 8–10 and caudal appendages of male larva; (C) sterna 6–10 of male larva; (D) sterna 7–10 of female larva. Photo by Rodolfo Novelo-Gutiérrez.

Figure 6 Details of the morphology of Perigomphus basicornis, female imago.

(A) head, frontodorsal view; (B) head, dorsal view; (C) right forewing, dorsal view; (D) right hindwing, dorsal view; (E) abdominal segments 8–10 and cerci, dorsal view; (F) ventral view of vulvar lamina and caudal appendages. Photo by Rodolfo Novelo-Gutiérrez.

Figure 7 Details of the morphology of Perigomphus pallidistylus larva.

(A) sterna 3–10 of male larva; (B) labrum, ventral view, showing row of gray setae; (C) tergites 8–10 and caudal appendages of male larva; (D) antennae, dorsal view, showing small holes on 3rd antennomeres. Photo by Rodolfo Novelo-Gutiérrez.

Thorax (Fig. 2A): wider than head, setose on inferior margin of pleura. Anterior margin of pronotum straight, lateral margins rounded, posterior margin wavy. Pronotal disk granulose; a light brown, large, irregular, glabrous area on each side of midline. Posterolateral margin of propleura bulging. Anterior and posterior wing pads parallel, reaching posterior margin of S4, with light and dark areas without a regular pattern, anterior wing pads lighter. Legs granulose (Fig. 2), short (e.g.: hind legs, when fully extended, reaching posterior margin of S8); fore- and middle femora short and stout, dorsolateral margins with a tuft of long, stiff setae (Fig. 5A); hind femora cylindrical, slightly compressed laterally, with a row of long, stiff setae on dorsal and ventral margins; surfaces of fore- and middle tibiae covered with stout, conic tubercles (Fig. 5A) and long setae, ending apically in a well-developed burrowing hook; a subapical, short brush of short, stiff setae on ventral surface of protibiae; hind tibiae cylindrical, slightly compressed laterally, with a ventral row of long, stiff setae, distal margins spiny; tarsi pale, covered with some long, stiff setae on dorsum, fore- and middle tarsi with short tubercles on ventral surface, hind tarsi with rows of short, spiniform setae; tarsal claws simple, with a pulvilliform empodium.

Abdomen (Fig. 2): fusiform, 1.7 times longer than its widest part, gently tapering caudad, brown on dorsum, yellowish-brown ventrally. S1–7 lacking dorsal protuberances, S8–9 with a very small, low protuberance on middle of posterior margin (Fig. 2A, Fig. 5B). Lateral margins of S2–3 slightly convex, straight on S4–7, slightly concave on S8–9; lateral margins of S2–6 with setae close to anterior margin, S7–9 without setae. All tergites with ill-defined color pattern, with small bare spots. Lateral spines on S6–9, reduced on S6, short on S7–9, tips rounded, increasing in size rearward (Fig. 2, Figs. 5C–5D). Sterna granulose (Fig. 5C–Fig. 5D), sterna 2, 5–7 and 9 divided into three sternites, sterna 3–4 and 8 divided into five sclerites (Fig. 2B, Figs. 5C–5D); sutures on sterna 2–3 slightly convergent posteriorly, 4–6 parallel, divergent caudally on 7–9 (Fig. 2B). Male gonapophyses absent, (Fig. 5A), female gonapophyses highly reduced (Fig. 5B), digitiform, roundly pointed, convergent apically. Caudal appendages granulose (Fig. 5B), twice longer than tergite 10. Epiproct triangular, male’s epiproct with two dorsal tubercles at basal 0.70 of its length (Fig. 5B), tip rounded. Cerci digitiform (Fig. 5B), acutely pointed. Paraprocts pyramidal (Figs. 5B–5D), roundly pointed, with a basal, transversal row of long, white, stiff setae (Figs. 5C–5D). Size proportions: epiproct 0.8, cerci 0.4, paraprocts 1.0.

Measurements. Exuviae (N = 2): TL 14.8–15.2 [15]; AL 9.0–9.1[9.05]; MwH 3.7–3.9 [3.8]; HfL 2.5; spine on S6 0.05, on S7 0.10, on S8 0.15, on S9 0.20. F-0 larvae (N = 8, mean in square brackets): TL 13.6–16.6 [15.1]; AL (ventral) 8.2–9.9 [9.0]; MwH 3.5–3.8 [3.7]; HfL 2.4–2.7 [2.5]; spine on S6 0.02–0.05 [0.04], on S7 0.05–0.10 [0.09], on S8 0.10–0.20 [0.13], on S9 0.15–0.24 [0.19].

Habitat. The larvae of P. basicornis were found inside a small seasonal creek (about 50 cm wide and 7 cm depth) and in the banks of second order streams, deep inside the forest. These water bodies are fast-flowing, pristine, surrounded by thick riparian vegetation providing shadow to the water course. Larvae preferred microhabitats with a bottom of pebbles, small rocks and coarse sand, pieces of fallen leaves and high dissolved oxygen concentrations (9.5 mg/L average) (Amaya-Vallejo, Novelo-Gutiérrez & Realpe, 2017).

Description of female imago: Head (Fig. 6A). Eyes yellowish-brown, with pale yellow borders; labium cream colored, dark brown medially, with long pale setae; labrum cream colored on distal half, brown on basal half; anteclypeus and postclypeus dark brown, postclypeus with three white spots, one on each side and one on the middle; frons dark brown with a large, transversal, dorsal white spot on each side of midline, not contiguous; dorsal and anterior surfaces of frons divided by a transversal, incomplete, border; vertex, occiput and rear of head dark brown. Numerous long, dark brown setae on the face and dorsum of head and along the occiput; posterior margin of occiput, in frontal view (Fig. 6A) wavy, in dorsal view (Fig. 6B) strongly concave and irregular with a large lobe to each side of midline; antenna dark brown (Fig. 6).

Thorax. Prothorax mostly light brown, lateral lobes with pale yellow wide spots covered with long black setae. Synthorax with mid-dorsal carina brown, mesepisternum brown with two pale stripes, one dorsal and connected to pale collar anteriorly, not reaching the antealar crest posteriorly, the second one running parallel to humeral suture but only on distal 55% of the length of mesepisternum, widened at distal end, as described for the male (Amaya-Vallejo, Novelo-Gutiérrez & Realpe, 2017); mesepimeron brown with a wide pale stripe full-length; metepisternum and metepimeron mostly pale with a fade brown stripe on metapleural suture, venter of pterothorax pale yellow. Coxae grayish yellow, prothoracic femora mostly dark brown, mesothoracic and metathoracic femora pale yellow, tibiae light to dark brown, tarsi and pretarsal claws blackish brown with distinct supplementary tooth; methatoracic femora with sturdy, thick black spines, the distal ones about as long as space between bases of adjacent spines and about as wide as femur width. Wings clear, with a slight yellow basal wash at bases. Venation dark brown; width of hind wing almost 1/3 its length. Second primary antenodal crossvein 7th in FW, 6th in Hw. Antenodal crossveins: Fw 14; Hw 8. Postnodal crossveins: Fw 12 (left), 11 (right); Hw 10 (left), 10 (right). Pterostigma gray, covering about 3.8 cells in Fw, 4.6 cells in Hw.

Abdomen. S1–8 slender, uniform in width. S1 pale yellow, S2-S4 blackish brown with narrow pale yellow mid-dorsal and lateral stripes; S5–7 with similar pattern but the mid-dorsal stripe shortens into a triangle reaching less than half the segment, and the lateral stripes appear discontinuous; S8-S10 all black (Fig. 6C). Cerci pale, yellow, almost doubling the size of S10, digitiform and parallel to each other, with short black and yellow bristles along entire length and slightly overpassing the epiproct, which is dark brown, rounded at the distal end and covered with long, black bristles (Fig. 6C). Paraprocts brown, round-shaped, extending half the length of cerci and covered with long, slender black setae. Vulvar lamina very wide at base, almost covering all the base of S8 and bulb-shaped, lobes also wide, with a width/length proportion of 0.71:1 and divergent tips (Fig. 6D).

Measurements (mm). TL 35.0, AL 24.1, FwL 24.0, HwL 23.0, HwW at nodus 8.1, Fw Pt 3.6, Hw Pt 3.9, MWh 6.0, HfL 5.0, CL 1.10.

Habitat. The female captured emerging in situ during the September 2016 field trip used the surface of a big round stone to emerge. The specimens reared under laboratory conditions, in an artificial habitat emulating the natural environment, preferred the sides of stones instead sticks or leaves. Then it is inferred that P. basicornis require stones protruding from water for emergence. As adults inhabit the canopy, riparian vegetation is of critical importance for their surviving and permanence.

Perigomphus pallidistylus (Belle, 1972), new record	
(Figs. 7A–7D)	

Material. Two F-0 larvae (male and female), one probably F-3 larvae (female). COLOMBIA: Department of Caldas; Municipality of Norcasia, Río Manso, elevation 672 m (5.6093667N, −74.95555W), 17 March 2016 (F-0 female), R.W. Sites leg., among marginal vegetation and rocks above dam. Same data but: Río Las Pavas, 664 m (5.5867N, −74.89215W), 18 March 2016 (F-0 male), R.W. Sites leg., rocky cobble; Quebrada Santa Rita, 544 m (5.6174833N, −74.910033W), 18 March 2016 (F-3 female), R.W. Sites leg., rock/gravel cascade.

This is the first published record of P. pallidistylus for Colombia, increasing southward its range extension, as it was previously recorded from the Pacific slope of Costa Rica to Canal Zone of Panama. (Garrison, Ellenrieder & Louton, 2006). Perigomphus pallidistylus was known from only five locations before, with an extent of occurrence less than 20,000 km2. Some of the known range area has been deforested, causing it to be listed inside the Vulnerable Category of the Odonata Red List (IUCN, 2018).

Discussion

After 28 years, the second larva and female for the genus are described; the larva and female of P. angularis still remain to be discovered. The behavior of adults difficult their capture, keeping the number of described species very low; raising the larvae under lab conditions seems to be an effective approach to increase the knowledge about the genus, being the rearing method presented here effective and easy to replicate.

Some additional considerations are important: larvae are very elusive and depict thanatosis when disturbed (VA Amaya-Vallejo, pers. obs., 2016), refusing to eat if they are fed directly, so living prey has to be put inside the habitat for them to eat when unobserved. After feeding trials with Artemia salina, Diptera (Chironomidae and Culicidae) larvae and Oligochaeta, the prey of choice was Grindall worms because they sink and stay in the bottom, where they are easily found and captured. Also, as observed from in situ and lab behavior, P. basicornis won’t use sticks or the submerged vegetation to emerge: they prefer rock surfaces, so it’s essential to make them available inside the artificial habitat (Fig. 8)

Figure 8 Details of the artificial habitat, with an emerging specimen of P. basicornis on a rock.

Photo by Vanessa Amaya-Vallejo.

The larvae of P. basicornis and P. pallidistylus can be differentiated by the following features (those of P. pallidistylus in parentheses): Anterior border of labrum with a dense brush of long, whitish, stiff setae (anterior border of labrum with a row of grayish-brown, setae ); third antennal segment lacking small, white holes (with numerous, small, white holes on dorsal surface); prementum with a maximum width-length ratio of 0.96:1 (maximum width-length ratio of 0.84:1); S8–9 with small, low, mound-like dorsal protuberances (no dorsal protuberances on S8–9); sternum 8 divided into five sclerites (sternum 8 divided into three sclerites); male’s epiproct with dorsal tubercles at basal 0.70 its length (male’s epiproct with dorsal tubercles at basal 0.50 its length); tips of male cerci not reaching the epiproct’s tubercles (tips of male cerci reaching the epiproct’s tubercles).

Key to the known F-0 larvae of Perigomphus

1. Sternum 8 divided into three sclerites (Fig. 7A); anterior border of labrum with a row of grayish-brown, setae (Fig. 7B); no dorsal protuberances on S8–9 (Fig. 7C); male epiproct little longer than its basal width, with dorsal tubercles at basal 0.50 the length of epiproct, tips of cerci reaching such tubercles (Fig. 7C); 3rd antennal segment with numerous, small, white holes on dorsal surface (Fig. 7B) .……..................................... pallidistylus

- Sternum 8 divided into five sclerites (Fig. 2B, Figs. 5C–5D); anterior border of labrum with a brush of white setae (Fig. 3A); small, low, mound-like dorsal protuberances on S8–9 (Fig. 2A, Fig. 5B); male epiproct as long as its basal width, with dorsal tubercles at basal 0.70 the length of epiproct, tips of cerci not reaching such tubercles (Fig. 5B); 3rd antennal segment lacking white holes on dorsal surface (Fig. 3B) …...............………. basicornis

Key to the known females of Perigomphus

1. Lobes of vulvar lamina narrow, parallel, in a width/length proportion of 0.38:1	pallidistylus*	
- Lobes of vulvar lamina wide, divergent, in a width/length proportion of 0.71:1 (Fig. 6D)	basicornis	

* Note: Females of P. pallidistylus were not available for comparison. The key was built based on the Westfall’s (1989, Fig. 3) drawing.

Conclusions

Perigomphus is still a poorly known genus. Based upon the recent new species described (Tennessen, 2011; Amaya-Vallejo, Novelo-Gutiérrez & Realpe, 2017), there is a high potential for more new species, mainly in Primary Neotropical Rainforests of South America. Now, two out of three larvae are described, enriching the knowledge about the genera.

Perigomphus could be considered as vulnerable because of its preference for pristine habitats which are continuously degraded, although searches for some more species are needed, in order to create phylogenetic and population genetics approaches; also, sampling for other populations or more specimens of P. basicornis is necessary to generate distribution models and genetic barcodes as complimentary identification tools. The method of rearing is effective although emergence of adults could take a substantial amount of time due to the particular behavior of the species.

Special thanks are due to Prof. Robert W. Sites for the donation of larvae of P. pallidistylus from Colombia. Dr. José Antonio Gómez-Anaya edited the larval images. The LSID from Amaya-Vallejo, Novelo-Gutiérrez & Realpe (2017) is urn:lsid:zoobank.org:pub:49E66BB1-2D0C-4BE0-BD0C- 067D1BEBB5EE.

Additional Information and Declarations

Competing Interests

Author Contributions

Field Study Permissions

Data Availability

The authors declare there are no competing interests.

Vanessa Amaya-Vallejo conceived and designed the experiments, performed the experiments, analyzed the data, authored or reviewed drafts of the paper.

Rodolfo Novelo-Gutiérrez analyzed the data, contributed reagents/materials/analysis tools, prepared figures and/or tables, authored or reviewed drafts of the paper, approved the final draft.

Emilio Realpe contributed reagents/materials/analysis tools, authored or reviewed drafts of the paper, approved the final draft.

The following information was supplied relating to field study approvals (i.e., approving body and any reference numbers):

Field research was approved by Parques Naturales Nacionales de Colombia office, Permit Number 005 of 2016.

The following information was supplied regarding data availability:

Material is stored in Universidad de Los Andes Natural History Museum (ANDES), Colombia and in Entomologic Collection of Instituto de Ecología de Xalapa (IEXA), México.

The research in this article did not generate any data or code. Images of the specimen are included in the article.

Material is stored in Universidad de Los Andes Natural History Museum (ANDES), Colombia and in Entomologic Collection of Instituto de Ecología de Xalapa (IEXA), México.

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
