# Peer review of "The larva and female of Perigomphus basicornis Amaya-Vallejo, Novelo-Gutiérrez & Realpe, 2017, and the first record of Perigomphus pallidistylus (Belle, 1972) for Colombia (Insecta: Odonata: Gomphidae)"

_PeerJ, doi:10.7717/peerj.5279_

## Round 0.1 · original submission · Minor Revisions

Please, revise your manuscript according to the suggestions given by the reviewers. Special attention is needed regarding methodology improvements.

·

Basic reporting

Please share pictures of the adult habitus (including the wings).
Include, if possible, an inventory number for your specimens.

Experimental design

no comment

Validity of the findings

no comment

Additional comments

This is a wonderful work which I really enjoyed to read. The photographs are also well done.
But please consider that this work can be improved by sharing sharp pictures of the wing venation. In paleontology (my primary field), we are dependent on the access to sharp images or drawings of the wing venation for comparing fossil and extant species. Please illustrate all four wings of your adult female (you mentioned intra-individual differences). This will also increase the impact of your study.

Reviewer 2 ·

Basic reporting

English language generally is very good, with a few inconsistencies; for example, ln. 35 'endemic to tropical' (not 'endemic of...'); ln. 46-47 use of terms like 'some' is vague. Why not provide the numbers of collected larvae and the number of moralities? ln. 48 '... one turned out to be..' is a bit journalistic '...one was a female.' would be better. ln. 279 'we reach 66% of the larvae for the genus' is not clear.

Cited literature is rather sparce and only briefly mentions Perigomphus. I would like to see a few more lines in the Introduction, and at the start of the Abstract, about the Gomphidae: what mates this a special odonate family (distributions, habitats, species and threats)? Please provide more context for the need for species diversity, conservation and the need for more taxonomic descriptions in Gomphids. Ln. 223 - lacks context. Please expand literature to indicate HOW far south the record of this species is (in km).

Detailed descriptions of larvae and imagos seem reasonable, and the figures are generally well presented. Please add scale bars to the figures.

Report is discovery driven and has no hypotheses as such. The manuscript is unfocussed as it provides (1) a species description (2) and a new record (P. pallidistylus) for Columbia. It is not clear why the authors' 2017 description (in Zootaxa) of MALE P. basicornis did not contain these descriptions of the FEMALES. Providing separate descriptions of sexes of the same species seems unhelpful for taxonomists.

Experimental design

Project describes odonate species, which I believe to be in the remit of PeerJ (as Biological Sciences), though I would expect species' descriptions to be in Zootaxa (which the authors did for the males of P. basicornis (see above).

There are no well-defined research questions: project is 'discovery based'. Description of species and records of new occurrences are information useful for conservation. However, the details of the fieldwork are very sparce. If there were robust field expeditions, then it would be useful to provide a map/list of field sites, and the dates of expeditions, to show areas where Perigomphus were NOT found. Can the authors provide more detailed information about the possible range of these species in Columbia? Or was there really only a single sampling trip?

Methods about the fieldwork are inadequate (as stated above). For example, ln. 161 - how did you arrive at the conclusions about microhabitat preference? What were the available microhabitats in the streams? What habitats in sampled streams was this species absent? What odonates co-occurred with these species? How was larval sampling performed?

Descriptions of exuviae and imago follow guidelines and are adequate.

Validity of the findings

Data appear robust. It would be nice to see scale bars on figures, and a link to the original images of different specimens so other researchers could measure these. No statistical analyses as such.

This manuscript is simply a species description (but expanding a previous description) and 1st record and there are few real conclusions that link to any question.

Can the authors include more data on specific habitat requirements (e.g. water temperature, surrounding vegetation, substrate type, pH) and co-occurring species? In a revision of this manuscript, there needs to be some indication about the type and number of sites where the authors (or collaborators) have searched for this P. basicornis (and failed to find it) as these data help understand its rarity/range. Also, there is some indication that P. basicornis requires pristine habitats (ln. 280), but there is no context about how the authors arrived at this conclusion.

Discussion about 'genetic barcodes' is weak - what marker would be used? I urge the authors to try and include some sequence data in a revision of this manuscript (it really is not costly to sequence 4-6 specimens) to understand the phylogeny of this genus.

There should be some discussion about the southward expansion of P. pallidystylus. Is this southern record due to range expansion or because of a prior lack of sampling for tropical odonates, for example? If an expansion is likely, then why and what is the possible route and where should this species be found?

Additional comments

Comments were made above.

---

## Round 0.2 · Minor Revisions

PeerJ has recently clarified its policies for manuscripts which report range extensions, life history information, or the description of new species, such that they should address a biological question or hypothesis as the focus of the submission and should be supported by multiple lines of evidence (e.g. morphology, genetics, etc.). I believe, however, that your manuscript could be brought in line with this policy via some additional edits and data provision. I note that you present "a protocol for larvae rearing" but, having looked more closely, this seems to be simply a table entitled "Standardized variables for Perigomphus basicornis rearing under laboratory conditions" listing the water temperature and CO2, air temperature and humidity, and the type of substrate. Please, can you elaborate on the process by which you optimized the conditions, and include the experimental data you generated. Doing so should help satisfy the requirement for answering a biological question(s).

---

## Round 0.3 · accepted · Accept

The mansucript has been improved and now can be accepted for publications. Thank you for choosing this Journal.

#